# Effects of Oral Bicarbonate Supplementation on the Cardiovascular Risk Factors and Serum Nutritional Markers in Non-Dialysed Chronic Kidney Disease Patients

**DOI:** 10.3390/medicina58040518

**Published:** 2022-04-05

**Authors:** Katarzyna Szczecińska, Małgorzata Wajdlich, Maja Nowicka, Michał Nowicki, Ilona Kurnatowska

**Affiliations:** 1Department of Internal Medicine and Transplant Nephrology, Medical University of Łódź, 90-153 Lodz, Poland; maja.a.nowicka@gmail.com (M.N.); ilona.kurnatowska@umed.lodz.pl (I.K.); 2Department of Nephrology, Hypertension and Kidney Transplantation, Medical University of Łódź, 92-213 Lodz, Poland; mwajdlich@gmail.com (M.W.); nefro@wp.pl (M.N.)

**Keywords:** metabolic acidosis, bicarbonate, chronic kidney disease

## Abstract

Background and Objectives: Kidneys play a key role in maintaining the acid–base balance. The aim of this study was to evaluate the effect of a 3-month oral sodium bicarbonate administration on arterial wall stiffness, arterial pressure and serum nutritional markers in non-dialysed patients with chronic kidney disease (CKD) stages 3–5 and metabolic acidosis. Methods: Eighteen CKD patients with eGFR < 45 mL/min/1.73 m^2^ and capillary blood bicarbonate (HCO_3_) < 22 mmol/L were enrolled in this single-centre, prospective study. Anthropometric parameters, pulse wave velocity, 24-h ambulatory blood pressure measurements, blood and urine parameters were assessed at the beginning and at the end of the study. The patients received supplementation with 2 g of sodium bicarbonate daily for three months. Results: A significant increase of pH: 7.32 ± 0.06 to 7.36 ± 0.06; *p* = 0.025, HCO_3_ from 18.7 mmol/L (17.7–21.3) to 22.2 mmol/L (20.2–23.9); *p* < 0.001 and a decrease in base excess from −6.0 ± 2.4 to −1.9 ± 3.1 mmol/L; *p* < 0.001 were found. An increase in serum total protein from 62.7 ± 6.9 to 65.8 ± 6.2; *p* < 0.013 and albumin from 37.3 ± 5.4 to 39.4 ± 4.8; *p* < 0.037 but, also, NT-pro-BNP (N-Terminal Pro-B-Type Natriuretic Peptide) from 794.7 (291.2–1819.0) to 1247.10 (384.7–4545.0); *p* < 0.006, CRP(C Reactive Protein) from 1.3 (0.7–2.9) to 2.8 (1.1–3.1); *p* < 0.025 and PTH (parathyroid hormone) from 21.5 ± 13.7 to 27.01 ± 16.3; *p* < 0.006 were observed, as well as an increase in erythrocyte count from 3.4 ± 0.6 to 3.6 ± 0.6; *p* < 0.004, haemoglobin from 10.2 ± 2.0 to 11.00 ± 1.7; *p* < 0.006 and haematocrit from 31.6 ± 6.00 to 33.6 ± 4.8; *p* < 0.009. The mean eGFR during sodium bicarbonate administration did not change significantly: There were no significant differences in pulse wave velocity or in the systolic and diastolic BP values. Conclusion: The administration of sodium bicarbonate in non-dialysed CKD patients in stages 3–5 improves the parameters of metabolic acidosis and serum nutritional markers; however, it does not affect the blood pressure and vascular stiffness.

## 1. Introduction

Metabolic acidosis is defined as a decrease of the plasma or venous blood bicarbonate (HCO_3_) level below 22 mmol/L [1]. The kidneys play a key role in maintaining the acid–base balance. As chronic kidney disease (CKD) progresses and the number of functioning nephrons decreases, metabolic acidosis develops. The acid–base balance is maintained by renal mechanisms—bicarbonate reabsorption and regeneration. The regeneration process comprises reabsorption and titratable acidity formation. The production of ammonium ion accounts for two-thirds of the bicarbonate regenerated in the kidney [1]. A loss of more than 80% of the functioning nephrons affects bicarbonate regeneration through ammoniogenesis impairment [2,3]. The development of metabolic acidosis correlates with a decreased glomerular filtration rate (eGFR), and this complication is estimated to occur in 30–50% of patients with eGFR < 30 mL/min/1.73 m^2^ [4,5,6]. It may also appear in earlier stages of CKD, especially if accompanied by tubular dysfunction, as in hyporeninaemic hypoaldosteronism occurring in diabetes [7]. Adverse consequences of metabolic acidosis include electrolyte disturbances, increased osteolysis, skeletal muscle atrophy, decreased protein synthesis, appetite loss, malnutrition, increased inflammation, impaired glucose tolerance and increased β2-microglobulin synthesis [8,9]. The results of experimental studies showed that a decline in GFR is mediated by metabolic acidosis [10,11]. Clinical observations show that metabolic acidosis accelerates CKD progression [6,12,13]. In the remaining nephrons, ammoniogenesis is increased and angiotensin II and endothelin are stimulated, resulting in progressive damage of the renal interstitium [14]. In turn, both metabolic acidosis and kidney failure accelerate the development of cardiovascular complications [15,16].

Increasing acid–base disturbances are a factor of poor prognosis; however, it should be noted that this relationship takes the shape of a “U” curve, as an increased risk of death occurs both at HCO_3_ concentrations below 22 mmol/L and above 32 mmol/L [17,18]. The suggested management at HCO_3_ ≤ 22 mmol/L in patients with CKD is the administration of oral sodium bicarbonate [18,19]. However, alkali therapy with sodium bicarbonate may increase the sodium load, which, in view of the impaired natriuresis in patients with CKD, may contribute to increased fluid accumulation and worsening of the arterial blood pressure (BP) control [20]. The data on the effect of sodium bicarbonate administration on BP are inconclusive [21,22,23].

The aim of the study was to evaluate the effects of a 3-month sodium bicarbonate administration on arterial wall stiffness, arterial blood pressure and serum nutritional markers in non-dialysed patients with CKD stages 3–5 with metabolic acidosis.

## 2. Materials and Methods

Patients with CKD diagnosed for at least 6 months with eGFR < 45 mL/min/1.73 m^2^ and HCO_3_ < 22 mmol/L were enrolled in this single-centre, prospective study. The exclusion criteria were as follows: current or past history of renal replacement therapy, active infection, unstable ischaemic heart disease and heart failure (CHF) in NYHA (New York Heart Association) classes III and IV, poorly controlled hypertension (BP < 160/90 mmHg) and the use of sodium bicarbonate or calcium carbonate.

During the screening visit, the patients were informed about the aims of the study, and after obtaining their informed consent and confirmation of the inclusion and exclusion criteria, they were qualified for the study. At the same visit, a detailed history was taken concerning the cause of CKD and used medications, especially antihypertensive drugs. Patients also underwent 24-h ambulatory blood pressure monitoring (ABPM). On the second visit, anthropometric data and pulse wave velocity measurements using SphygmoCor were taken. At the same time, fasting blood samples for a complete blood count were collected; biochemical serum parameters: creatinine, sodium, potassium, calcium, phosphate, parathyroid hormone (PTH), albumin, total protein, glucose, NT-proBNP (N-Terminal Pro-B-Type Natriuretic Peptide) and CRP (C Reactive Protein) were determined, and capillary blood gas tests were done, as well as the measurements of the concentrations of creatinine, sodium and albumin in the first morning void urine sample. The eGFR was estimated with the CKD-EPI formula. On this visit, 120 self-made cachets with 2 g of sodium bicarbonate were given to the patient with instructions to take one once daily before the first meal of the day. Three months later, the patients’ history of the last 3 months, including any adverse events, was taken, patients were asked to return the remaining cachets to assess their compliance and all the procedures from the second visit were repeated.

The study was initiated after the approval of the protocol by the local ethics committee (RNN/273/15/KE), and informed consent was obtained from each patient before the start of any study procedures.

Statistical analysis. Continuous data were presented as the means with standard deviations (SD) or as medians with interquartile ranges, depending on the normality of the distribution, which was tested with the Shapiro–Wilk test. Continuous data with normal distribution were compared between visits with paired Student’s *t*-tests and between patient groups with unpaired Student’s *t*-tests. Continuous non-normally distributed and ordinal data were compared between visits with Wilcoxon’s test and between patient groups with the Mann–Whitney *U* test. Nominal data were presented as numbers with percentages and compared with the chi-square test. *p*-values lower than 0.05 were considered statistically significant. All analyses were performed with Microsoft Excel (Microsoft, Baton Rouge, LA, USA) or with Statistica 13.3 software (Dell, Round Rock, TX, USA).

## 3. Results

Thirty-two patients of Caucasian origin, all treated chronically at one outpatient nephrological clinic in Central Poland, were enrolled to the study. The final analysis included the results of 18 patients (12 M and 6 F) at a mean age of 66.1 ± 11.7 years and with eGFR 19.3 ± 9.2 mL/min/1.73 m^2^. Ten patients did not attend the second visit at the appointed time for personal reasons, including fear of COVID-19 infection; four patients were excluded due to noncompliance during the 3-month bicarbonate administration period. The causes of CKD among the patients who finished the study were: glomerular disease (four patients), diabetic kidney disease (two patients), hypertensive kidney disease (three patients), polycystic kidney disease (two patients), obstructive uropathy (three patients) and unknown (four patients). The patients were receiving antihypertensive drugs, including angiotensin receptor antagonists (4 patients), angiotensin-converting enzyme inhibitors (3 patients), β-blockers (11 patients), calcium channel blockers (11 patients), α-blockers (7 patients), α2-adrenergic receptor agonists (1 patient), loop diuretics (10 patients), aldosterone antagonists (2 patients) and thiazide-like diuretics (1 patient). The anthropometric data of the analysed patients are presented in Table 1.

The 3-month administration of sodium bicarbonate significantly reduced the metabolic acidosis parameters (Figure 1A–C). There was an increase in the pH (7.32 ± 0.06 vs. 7.36 ± 0.06; *p* = 0.025, Figure 1A) and HCO_3_ (18.7 mmol/L (17.7–21.3) vs. 22.2 mmol/L (20.2–23.9); *p* < 0.001, Figure 1B) and decrease in the base excess (−6.0 ± 2.4 vs. −1.9 ± 3.1mmol/L; *p* < 0.001, Figure 1C). When analysing the changes in the HCO_3_ concentrations in individual patients, only one patient showed no change of this parameter. Individual changes of pH are shown in Figure 2.

The mean eGFR during sodium bicarbonate administration did not change significantly: 19.3 ± 9.2 mL/min/1.73 m^2^ vs. 18.7 ± 10.0 mL/min/1.73 m^2^; *p* = 0.360. There was no effect of sodium bicarbonate administration on the urea, sodium, potassium, calcium and phosphate concentrations (Table 2). On the other hand, increased serum total protein and albumin concentrations but also NT-pro-BNP, CRP and PTH levels were observed, as well as an increase in the erythrocyte count, haemoglobin and haematocrit. There was no effect of bicarbonate administration on the urinary sodium concentration, but an ACR increase was observed during drug administration. Detailed data are presented in Table 2.

No difference in the pulse wave velocity was observed after the sodium bicarbonate intake period. There was no significant difference in the systolic and diastolic BP values, both during the day and night. Some patients (38%) required an increase in the doses of antihypertensive drugs, and 16.6% of patients required an increase in the doses of diuretics. Detailed data are presented in Table 3.

## 4. Discussion

Our prospective study among non-dialysed CKD patients showed that sodium bicarbonate administration significantly reduced the metabolic acidosis parameters but had no effect on kidney function. The data from various studies, including experimental models, have been inconclusive [21,22,24,25]. In experimental studies, the potential of sodium bicarbonate supplementation and an alkalising diet to slow the progression of kidney failure, as expressed by a decrease of eGFR, has been demonstrated [10,26]. Small, single-centre clinical trials appear to confirm the nephroprotective effect of alkalinisation. The administration of sodium bicarbonate slows the progression of kidney disease, may delay the implementation of renal replacement therapy and improve the indicators of nutritional status [2]. The effect of delaying the progression of kidney failure as a result of sodium bicarbonate administration has been observed in studies among patients with hypertensive nephropathy in the early stages of CKD without metabolic acidosis [22], as well as among patients with advanced CKD and known metabolic acidosis [21,24].

In a recently published study with a mean follow-up period of 1.35 ± 0.75 years, no effect of bicarbonate administration on the progression of kidney failure was observed [25], similar to our study. The amount of bicarbonate administered in our trial was also similar (2.6 g vs. 2.0 g in our study). In the studies in which the effect of bicarbonate administration on kidney failure progression was observed, the doses of administered bicarbonate were higher than in our study, and the study periods were longer. Mahajan et al. used a dose of 0.5 mEq/kg of lean body mass [22]; de Brito-Ashurst et al. used a dose of 1.8 g and titrated the dose of sodium bicarbonate to achieve a HCO_3_ ≥ 23 mEq/L [21]. In those studies, diverse causes of CKD were highlighted, as in our study, whereas the studies that demonstrated a positive effect of sodium bicarbonate supplementation on kidney function usually recruited patients with hypertension nephropathy [22]. Furthermore, in the discussed studies, a reduction in potassium concentration was observed in the group of patients receiving bicarbonate [25]; a similar, but not statistically significant, trend was observed in our study.

Hypertension is an important modifiable risk factor for CKD progression and one of the major risk factors for cardiovascular disease [27]. ABPM proved to be a more valuable predictor of cardiovascular disease and mortality than office blood pressure measurements [28]. In theory, the administration of sodium bicarbonate could result in the worsening of BP control because of the excessive sodium load (1-g sodium bicarbonate contains about 12-mmol Na^+^). In hypertensive patients, especially those with CKD, a low-sodium diet is recommended [29]. In our study, one-third of the patients required a minor adjustment of antihypertensive treatment, including an increase in the diuretics dose, but finally, we did not observe any significant deterioration in the hypertension control. According to ABPM, both systolic and diastolic BP were not significantly altered during the administration of sodium bicarbonate, either during the day or at night. Similarly, no such correlation was found in the study by Melamed et al., although, in their study, some patients also required an increase in antihypertensive medications (19% in the treated group vs. 9% in the placebo group), including the use of diuretics (8% vs. 5%) [25].

In their study, patients treated with alkali therapy were more likely to have an exacerbation of heart failure (CHF), as measured by the number of hospitalisations [25]. In our study group, hospitalisation for symptoms of CHF exacerbation was not required in any case, but the NT-proBNP level was objectively found to be increased. In the BiCARB study, no effect of sodium bicarbonate administration on the NT-proBNP levels in CKD patients was found [30].

Sodium bicarbonate supplementation could theoretically cause effects similar to those observed with sodium chloride (NaCl) administration, such as oedema, increased body weight or increased symptoms of CHF, as described in other studies [31,32,33,34].

A study by Husted et al. comparing the effects of NaCl and sodium bicarbonate, in which sodium chloride was restricted in the subjects’ diets, found negative effects on the body weight and blood pressure only in the group taking NaCl [35]. However, in another study by Husted et al. in which dietary NaCl was not restricted, an increase in both body weight and blood pressure was observed in both groups [36].

Subsequent studies comparing sodium chloride and sodium bicarbonate supplementation, with NaCl restriction in the diet of both groups of subjects, showed some effects of both substances only as increases in body weight but not in BP; an increase in BP was observed only in subjects taking NaCl [37,38]. Sodium bicarbonate supplementation does not seem to have such a significant effect on body weight and oedema as sodium chloride administration, as confirmed by studies in which the supplementation of alkalising agents did not increase the body weight or BP [21,23,24,39,40].

Metabolic acidosis may be a risk factor for cardiovascular disease, since one of the effects of metabolic acidosis is deterioration of the myocardial function [41] and, also, activation of the RAA system and vascular endothelial dysfunction [3]. Dobre et al., found a nonlinear relation between the bicarbonate concentration and CHF [13]. Again, in the MESA study, higher bicarbonate concentrations were associated with increased arterial stiffness and left ventricular mass and an increased risk of CHF in the general population [42]. The preliminary results from the randomised, controlled, clinical SOBIC trial, which enrolled CKD patients in stages 3 and 4 and with HCO_3_ < 21 mmol/L, showed no worsening of BP control in patients undergoing sodium bicarbonate supplementation [43]. A lack of effect on BP control may be related to the reported increased natriuresis during sodium bicarbonate supplementation compared to natriuresis measured during sodium chloride administration [35], although, in our study, no increase in sodium excretion was observed during the drug intake. In the already mentioned studies analysing the effect of sodium bicarbonate administration on the progression of CKD [21,22], as in our study, no differences in BP were found between the study group and the control group. The exclusion criteria of these studies included advanced CHF and poorly controlled hypertension, as in our study. In a recently published study, patients with CKD stages 3 and 4 with concomitant chronic diseases (diabetes, hypertension and ischaemic heart disease) who were given sodium bicarbonate for 6 months required an increase in diuretic administration [23]. Two meta-analyses published in 2019 presenting studies that examined the effect of bicarbonate administration on CKD showed conflicting results regarding the effect of supplementation on BP control [44,45].

As mentioned earlier, the presence of metabolic acidosis is a risk factor for cardiovascular complications. It has been shown to cause inflammation of vascular endothelial cells [46], increase the endothelin-1 levels [47], which increases the vascular inflammation, fibrosis and extracellular matrix formation [48], which may be expressed as increased arterial wall stiffness, leading to a higher risk of CVD. An association between arterial stiffness, measured as the pulse wave velocity, and cardiovascular events has been reported in several studies investigating CKD populations [49,50,51]. Data on the effects of acid–base imbalance on arterial stiffness are conflicting. In animal models, it was shown that bicarbonate supplementation was associated with an increased risk of vascular calcification [52]. In an experimental study, metabolic acidosis was found to protect against renal calcification formation among animals on a phosphate-rich diet [53]. The MESA study found a relationship between higher bicarbonate concentrations and increased arterial stiffness and left ventricular mass among the general population [42]. There was no association observed between the bicarbonate concentration and arterial stiffness among subjects with CKD or those from a healthy population in the Health ABC study [54]. In contrast, the KNOW-CKD study found that lower bicarbonate concentrations were accompanied by higher arterial stiffness in patients with CKD [55]. An inverse correlation between the serum bicarbonate concentration and vascular stiffness was observed in haemodialysis patients [56,57]. In our study, the administration of sodium bicarbonate did not significantly affect vascular stiffness. Metabolic acidosis results in reduced albumin synthesis [58]. The NHANES II study, where over 1500 patients > 20 years old were included, found that the odds ratio for hypoalbuminaemia increased from 1.0 at HCO_3_ > 28 mmol/L to 1.54 at HCO_3_ ≤ 22 mEq/L [5]. In our study, sodium bicarbonate supplementation in our study resulted in an expressed increase in the serum total protein and albumin levels. No subject was on a special diet, vegetarian diet or protein-restricted (protein < 0.8g/kg/d), and changes in eating habits were not recommended during the whole study. None of the patients received nutritional complements or appetite stimulants. Metabolic acidosis in CKD is one of the factors causing PEM (protein–energy malnutrition), which is characterised among others by hypoalbuminaemia [9]. The mechanisms that may contribute to the development of PEM in metabolic acidosis include increased protein catabolism, decreased protein synthesis, insulin resistance and decreased leptin levels [59]. An effect similar to our observations was noted in a multicentre randomised trial in which increased albumin levels were observed after a 36-month period of bicarbonate supplementation [60]. Improvements in the nutritional status were also observed in patients undergoing sodium bicarbonate supplementation, expressed in increases in the albumin levels and arm muscle circumferences [21,61]. On the other hand, a study by Jiwon Jeong et al. found no improvement in the nutritional status/changes in the albumin concentration due to alkaline therapy in patients with CKD stages 4 and 5 [39]. In contrast, among pre-dialysis patients who received an average of 4.5 ± 1.5 g/d of sodium bicarbonate, a significant increase in serum albumin concentration was observed after a 6-month supplementation period [62].

The improvement of red cell parameters observed in our study group may be connected i.a. with a reduction of acidosis and improvement of the nutritional status. Animal experimental models suggest that metabolic acidosis among patients with CKD increases the concentration of hepcidin, which plays a role in the pathogenesis of anaemia in CKD [63,64]. Acidosis has been shown to increase the need for erythropoiesis-stimulating agents (ESAs) [65] (our patients were not treated with ESAs). We observed a significant increase in the haemoglobin concentration during the 3-month sodium bicarbonate supplementation, and the results were similar to de Brito-Ashurst et al.’s findings [21]. On the other hand, Ori et al. [66] found no effect of sodium bicarbonate supplementation on the haemoglobin concentration.

## 5. Limitations of the Study

The first and most important limitation of our study is the small group size, which may lower the generalisation of the study. However, this relatively small group and short intervention time allowed the close monitoring of patients and control of the compliance. The lack of a control group is another limitation. The placebo group was not planned, since the patients with HCO_3_ < 22 mmol/L, according to the 2012 KDIGO recommendations, should be receiving sodium bicarbonate [18]. As sodium intake is associated with elevated blood pressure and cardiovascular disease, the objective evaluation of consumed sodium is important. The measurements of 24-h urinary sodium excretion is the preferred method of estimating sodium intakes [67]; however, 24-h collections are expensive and relatively burdensome to individuals. Some data indicate that models based on sodium, potassium and creatinine concentrations from casual urine specimens may be useful for the prediction of 24-h sodium excretion [68]. We have not observed the changes in sodium excretion after NaHCO_3_ supplementation, but the lack of measurement of 24-h urinary sodium excretion is a limitation of our study. The present study is a pilot study, and the small group size is due to the narrow inclusion criteria. In the era of the COVID-19 pandemic, visits to both the hospital and the outpatient clinic were associated with patient anxiety and an increased risk of infection; therefore, a control group was not formed. A study with a larger sample size and longer follow-up period is planned.

## 6. Conclusions

The administration of sodium bicarbonate in non-dialysed CKD patients in stages 3–5 improves the parameters of metabolic acidosis and serum nutritional markers; however, it does not affect the blood pressure and vascular stiffness.

## Figures and Tables

**Figure 1 medicina-58-00518-f001:**
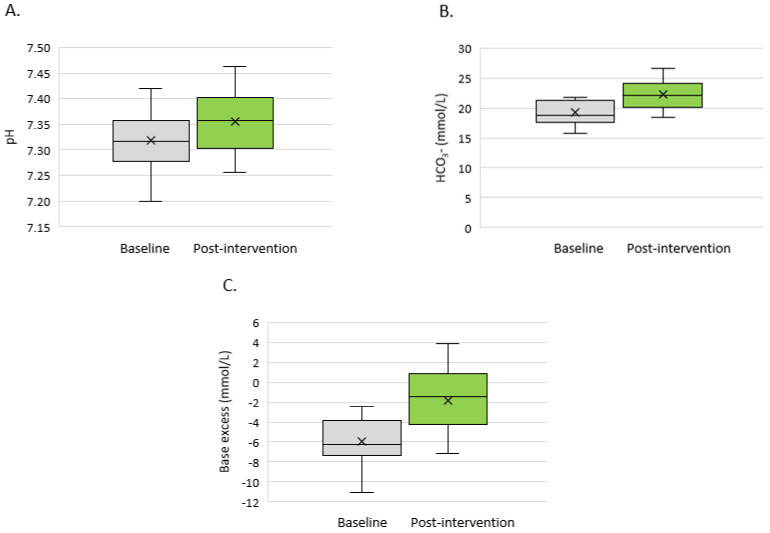
Acid–base balance parameters at the baseline visit and after a 3-month sodium bicarbonate administration in non-dialysed CKD 3–5 stages patients (*n* = 18): (**A**) pH, (**B**) HCO_3-_ (mmol/L) and (**C**) base excess (mmol/L).

**Figure 2 medicina-58-00518-f002:**
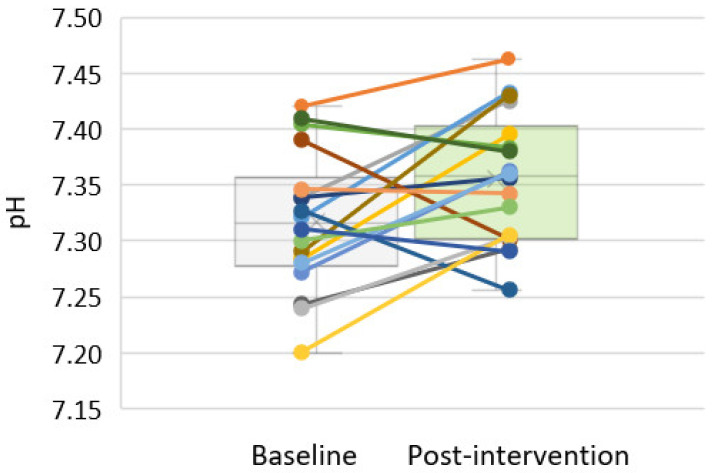
Individual blood pH changes between the baseline visit and after a 3-month sodium bicarbonate administration in non-dialysed CKD 3–5 stages patients (*n* = 18).

**Table 1 medicina-58-00518-t001:** Anthropometric parameters at the baseline visit and after a 3-month sodium bicarbonate administration in non-dialysed chronic kidney disease (CKD) 3–5 stages patients.

	Visit 0	Visit 1	*p*-Value
Sex (M/F)	66.6% (*n* = 12)/33.3% (*n* = 6)		-
Age (years)	66.08 (11.66)		-
Height (cm)	170.72 (8.60)		-
Weight (kg)	74.44 (15.96)	75.67 (16.12)	0.058
BMI (kg/m^2^)	23.83 (22.67–27.12)	25.11 (23.40–28.40)	0.051

Values are presented as the mean ± SD or median (IQR). Abbreviations: M—male, F—female and BMI—Body Mass Index.

**Table 2 medicina-58-00518-t002:** Serum biochemical and complete blood count parameters and urine biochemical parameters at the baseline visit and after a 3–month sodium bicarbonate administration in non–dialysed CKD 3–5 stages patients.

	Visit 0 (*n* = 18)	Visit 1 (*n* = 18)	*p*-Value
Creatinine (µmol/l)	313.09 (126.25)	333.66 (140.03)	0.057
eGFR (ml/min/1.73 m^2^)	19.33 (9.20)	18.67 (9.97)	0.360
Na (mmol/l)	138.29 (3.30)	139.48 (3.42)	0.151
K (mmol/l)	4.91 (0.59)	4.83 (0.55)	0.545
Ca (mmol/l)	2.22 (0.14)	2.25 (0.18)	0.317
P (mmol/l)	1.43 (0.25)	1.44 (0.33)	0.688
Albumin (g/l)	37.26 (5.43)	39.35 (4.81)	0.037 *
Total protein (g/l)	62.71 (6.92)	65.8 (6.22)	0.013 *
NT-proBNP (pmol/l)	794.70 (291.20–1819.00)	1247.10 (384.70–4545.00)	0.006 *
CRP (mg/l)	1.30 (0.70–2.93)	2.75 (1.10–3.10)	0.025 *
PTH (pmol/l)	21.54 (13.78)	27.07 (16.29)	0.006 *
RBC (10^6^/µl)	3.39 (0.60)	3.61 (0.57)	0.004 *
HGB (g/dl)	10.24 (1.95)	10.99 (1.72)	0.006 *
HCT (%)	31.64 (5.99)	33.55 (4.80)	0.009 *
ACR (mg/g)	479.60 (89.50–807.40)	872.40 (155.70–1412.80)	0.023 *
Urine Na/Cr (mmol/mmol)	16.52 (5.34)	18.24 (9.28)	0.467

Values are presented as the mean (SD) or median (IQR). Statistically significant *p*-values are marked with asterisks. Abbreviations: eGFR—estimated glomerular filtration rate, Na—sodium, K—potassium, Ca—calcium, P—phosphate, CRP—C–reactive protein, NT-pro-BNP—N-terminal-brain natriuretic prohormone, PTH—parathyroid hormone, RBC—red blood cell, HGB—haemoglobin, HCT—haematocrit, ACR—albumin to creatinine ratio and urine Na/Cr—urine sodium to creatinine ratio.

**Table 3 medicina-58-00518-t003:** Blood pressure and pulse wave velocity at the baseline visit and after a 3–month sodium administration in non–dialysed CKD 3–5 stages patients.

	Visit 0 (*n* = 18)	Visit 1 (*n* = 18)	*p*-Value
Dipper (yes/no)	41.2% (7)/58.8% (10)	41.2% (7)/58.8% (10)	1.000
SBP mean (mmHg)	133.00 (14.07)	138.06 (13.31)	0.304
DBP mean (mmHg)	75.75 (11.20)	80.00 (8.23)	0.077
HR mean (BPM)	73 (67–74)	74 (67–79)	0.070
SPB day (mmHg)	134.31 (13.58)	138.94 (14.63)	0.386
DBP day (mmHg)	77.13 (11.67)	81.38 (8.22)	0.108
MAP day (mmHg)	96.19 (11.18)	100.56 (8.67)	0.206
SBP night (mmHg)	129.06 (16.94)	134.94 (13.94)	0.168
DBP night (mmHg)	71.69 (11.61)	75.75 (11.42)	0.070
MAP night (mmHg)	90.37 (11.63)	89.86 (25.11)	0.930
NBPF (%)	5.53 (6.39)	4.72 (9.93)	0.679
PWV (m/s)	9.81 (3.02)	9.29 (3.78)	0.464

Values are presented as the mean ± SD or median (IQR). Abbreviations: SBP—systolic blood pressure, DBP—diastolic blood pressure, HR—heart rate, MAP—mean arterial pressure, NBPF—nocturnal blood pressure fall and PWV—pulse wave velocity. None of the patients reported any adverse effects associated with taking the bicarbonate. All patients wished to continue taking sodium bicarbonate after the study, stating a subjective improvement in well-being.

## Data Availability

Data are available from the corresponding author upon request.

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
