# Peer review of "Effects of Oral Bicarbonate Supplementation on the Cardiovascular Risk Factors and Serum Nutritional Markers in Non-Dialysed Chronic Kidney Disease Patients"

_medicina, 2022, doi:10.3390/medicina58040518_

Round 1
Reviewer 1 Report
Comments to the author:
In this manuscript entitled “Effects of oral bicarbonate supplementation on the cardiovascular risk factors and nutritional status in non-dialyzed chronic 3 kidney diseases patients”, the authors tried to demonstrate cardiovascular safety under oral bicarbonate supplementation in CKD patients. Although this is an important issue, some significant concerns are raised that require your attention before the article can be accepted. The major limitation of this research is its study design of a small sample size across a wide range of CKD stages during a short follow-up period.
Major concerns:
- About the title and their study aims, they used terms of “cardiovascular risk factors” and “nutritional status”. However, they only checked nutritional indicators without measuring anthropometric parameters, such as mid-arm muscle circumference or exercise endurance. They should also discuss more why using ambulatory blood pressure monitoring, pulse wave velocity, and NT-proBNP as cardiovascular risk factors.
- The authors should state specific objectives, including any prespecified hypotheses in the last paragraph of the "Introduction” section.
- There were only 18 participants across 3 stages of chronic kidney disease. We have known that the ability of renal handling of acid-base balance and sodium excretion is quite different if eGFR < 20 ml/min. The authors should demonstrate and discuss the differences.
- There is a strikingly high percentage of loop diuretics users (10/18 patients) in their study patients. They should describe the change of prescription on diuretics during the study period because this may influence the study results.
- Nutritional approaches are also helpful in treating metabolic acidosis. Low protein and eating fruits and vegetables containing alkali were proved to be effective. They should provide related data about the dietary intervention or management during the study period or discuss this.
- They did not observe a significant change in the data of urine Na/Cr. This may be related to the spot rather than 24-hr urine collection.
- The authors should also discuss the generalisability (external validity) of the study results according to the STROBE Statement for EBM.
- Finally, many sentences are verbose, so it is difficult to read. I strongly recommend a native English editor to revise the writing.
Author Response
Reviewer 1
Comment 1. About the title and their study aims, they used terms of “cardiovascular risk factors” and “nutritional status”. However, they only checked nutritional indicators without measuring anthropometric parameters, such as mid-arm muscle circumference or exercise endurance. They should also discuss more why using ambulatory blood pressure monitoring, pulse wave velocity, and NT-proBNP as cardiovascular risk factors.
Response: Thank you for your comment. Indeed we didn’t measure mid-arm circumference or exercise endurance, which would help better assess the nutritional status. We have changed the title of the article to „Effects of oral bicarbonate supplementation on the cardiovascular risk factors and serum nutritional markers in non-dialyzed chronic kidney diseases patients.”, in order to reflect our approach more accurately. We have also discussed blood pressure monitoring and pulse wave velocity as cardiovascular risk factors more thoroughly and added specific references. We did not discuss why we decided to assess NT-proBNP because it is acknowledged factor of heart failure.
Comment 2. The authors should state specific objectives, including any prespecified hypotheses in the last paragraph of the "Introduction” section.
Response: Thank you for your remark, Introduction section was revised according to your suggestions.
Comment 3. There were only 18 participants across 3 stages of chronic kidney disease. We have known that the ability of renal handling of acid-base balance and sodium excretion is quite different if eGFR < 20 ml/min. The authors should demonstrate and discuss the differences.
Response: Thank you for your comment, due to the small group size and since only three subjects among the participants had eGFR value exceeding 30 ml/min/1.73 m2, we decided not to differentiate subgroups.
Comment 4. There is a strikingly high percentage of loop diuretics users (10/18 patients) in their study patients. They should describe the change of prescription on diuretics during the study period because this may influence the study results.
Response: The dose of diuretic was increased only in 3 cases, we have included this information in the result section. The described changes were regarding 2 cases of loop diuretics and 1 case of thiazide-like diuretic.
Comment5. Nutritional approaches are also helpful in treating metabolic acidosis. Low protein and eating fruits and vegetables containing alkali were proved to be effective. They should provide related data about the dietary intervention or management during the study period or discuss this.
Response: Thank you for this remark, we have added the information that none of the participants was on vegetarian diet. We agree that diet rich in fruit and vegetables has alkalizing effect and it is benefficial, especially when it comes to patients with chronic kidney disease and metabolic acidosis. However, as we stated, no subject in our study was on a special diet and changes in eating habits were not recommended during the whole study.
Comment 6. They did not observe a significant change in the data of urine Na/Cr. This may be related to the spot rather than 24-hr urine collection.
Response: Of course, we agree that 24-hour urine collection is a more acurate method of estimating natriuresis, than spot urinary sodium. However spot urinary sodium method is more convenient for the patient and reduces the risk of non-compliance.
Comment 7. The authors should also discuss the generalisability (external validity) of the study results according to the STROBE Statement for EBM.
Response: Thank you for your remark, small sample and recruiting patients from one centre may cause lower generalisibility of the study, we have added this information in the limitations section. On the other hand, the fact that the participants have many comorbidities (only poorly controled hypertension and heart failure, stages- NYHA III and IV, excluded patients from the study), as well as no upper age limit, may contribute to higher applicability of the study.
Comment 8. Finally, many sentences are verbose, so it is difficult to read. I strongly recommend a native English editor to revise the writing.
Response: Thank you for that remark, the writing has been revised, according to the Reviewers suggestions.
Reviewer 2 Report
Line 67: Add power/alpha for the study.
Line 71: "(RR<160/90 mmHg)": I believe it should be BP* <160/90. Please correct it.
Line 139: Could it be because the follow-up period was too short (only 3 months)?
Line 141: Unnecessary use of the word "natrium" instead of "sodium".
-Authors have correctly acknowledged the limitations of this study. One of the major limitations of this study is the small sample size and short follow-up period. It is possible that with adequate sample size and a long follow-up period, the authors may see completely different results.
Author Response
Reviewer 2
Comment 1. Line 67: Add power/alpha for the study.
Thank you for your comment, we acknowledge the study is underpowered due to small sample size and its exploratory character. Sample size has been estimated to achieve the co-primary objectives of the study, which are the detection of changes in the average 24 hour systolic and diastolic blood pressure in ABPM at baseline visit and after a 3-month sodium bicarbonate administration in non-dialyzed CKD 3-5 stage patients. Based on screening results we estimated average systolic blood pressure in this population to be 130 (SD 15) mmHg and diastolic blood pressure to be 75 (SD 10) mmHg. On the assumption that we would like to detect a 15% change in blood pressure after a 3-month sodium bicarbonate administration, with Ro of 0.3, 90% power and alpha level of 0.05, a sample size of N=9 for change in diastolic blood pressure and N=11 for change in systolic blood pressure would be required. In fact the power of the study is lower, as the change of SBP and DBP is smaller than we assumed.
Comment 2. Line 71: "(RR<160/90 mmHg)": I believe it should be BP* <160/90. Please correct it.
Response: Thank you for your suggestion, we have applied the necessary corrections.
Comment 3. Line 139: Could it be because the follow-up period was too short (only 3 months)?
Response: Thank you for this comment, the follow-up period was designed to maximise compliance and at the same time enable to investigate the influence of sodium bicarbonate on blood pressure, pulse wave velocity and nutritional markers, we agree that its duration limits the results concerning kidney function.
Comment 4. Line 141: Unnecessary use of the word "natrium" instead of "sodium".
Response: Thank you for your suggestion, we have applied the necessary corrections.
Comment 5. -Authors have correctly acknowledged the limitations of this study. One of the major limitations of this study is the small sample size and short follow-up period. It is possible that with adequate sample size and a long follow-up period, the authors may see completely different results.
Response: Thank you for this remark, we agree that our study has some limitations, such as small sample size. Though the studies investigating the influence of natrium bicarbonate on blood pressure control, pulse wave velocity are sparse, the available results, mentioned in the discussion section, appear to be consistent with ours.
Reviewer 3 Report
- In the abstract, the squared symbol should be superscript.
- In the materials and methods section the number of patients that were screened and qualified should be mentioned. The center where the study takes place should also be mentioned.
- On p3 line 113, "4 patients" should be in parentheses.
- Table 1: it would be interested to include race as an anthropometric parameter if such data is available
- Figure 1 A - the y axis is cut off and the scale should start at 0. The x axis for all 3 graphs should be labeled. For B., the error bars are both plus and minus while A and C are plus only - all three graphs should be the same. N number should be included in the figure legend.
- For Figure 2, x axis should be labelled and n number should be included in figure legend.
- Table 2: Legend should include n number.
- p6 line 151 should read "statistically significant" and line 155 should read "creatinine ratio"
- Line 280: should read "others"
- Line 3015: the authors should not being a sentence with the word "But"
- Table 3 should have n number.
- p7 line 171 - you should clarify what treated conservatively means
Author Response
Comment1. In the abstract, the squared symbol should be superscript.
Response: Thank you for your suggestion, we have applied the necessary corrections.
Comment 2. In the materials and methods section the number of patients that were screened and qualified should be mentioned. The center where the study takes place should also be mentioned.
Response: Thank you for this comment, we stated the number of screened and qualified patients in results section, we also have added information about the centre, that the study took place.
Comment 3. On p3 line 113, "4 patients" should be in parentheses.
Response: Thank you for this comment, we have applied the necessary changes.
Comment 4. Table 1: it would be interested to include race as an anthropometric parameter if such data is available
Response: Thank you for your comment, all of the participants were Caucasian, we have added this information to the results section.
Comment 5. Figure 1 A - the y axis is cut off and the scale should start at 0. The x axis for all 3 graphs should be labeled. For B., the error bars are both plus and minus while A and C are plus only - all three graphs should be the same. N number should be included in the figure legend.
Response: Thank you for this comment, we added (n=18) to all figure legends and labeled x axis (and therefore removed the legend from the side). Moreover, to standardize the figures we decided to change them to boxplots. For plot A (pH) and C (Base excess) the axis do not start from 0 as in A it would not be proper for pH scale and in C the variables have both positive and negative values.
Comment 6. For Figure 2, x axis should be labelled and n number should be included in figure legend.
Response: Thank you for this remark, necessary changes have been applied.
Comment 7.Table 2: Legend should include n number.
Response: Thank you for this comment, the n number was added to the table 2.
Comment 8. p6 line 151 should read "statistically significant" and line 155 should read "creatinine ratio"
Response: Thank you for this suggestion, the necessary changes were applied.
Comment 9. Line 280: should read "others"
Response:Thank you for your suggestion, we have applied the necessary corrections.
Comment 10. Line 3015: the authors should not being a sentence with the word "But"
Response:Thank you for this remark, the necessary changes were applied.
Comment 11. Table 3 should have n number.
Response:Thank you for this comment, the n number was added to the table.
Comment 12. p7 line 171 - you should clarify what treated conservatively means
Response:Thank you for this comment, as by conservatively treated we mean non-dialyzed, we have used this term instead
Round 2
Reviewer 1 Report
I know the authors had tried their best to respond to my comments. However, as I have mentioned, the major weakness of their research is the study design of a small sample size across a wide range of CKD stages. I think the response to my comment 3 is not justified and insufficient to support their conclusions. The renal handling of acid-base balance and sodium excretion is significantly limited while eGFR less than 20 ml/min. They should provide detailed data (Cr, eGFR, Urea, Na, NT-proBNP, ACR, Urine Na/Cr, SBP, DBP, MAP, NBPF, PWV, HCO3-, and PH) at baseline and post-intervention among each patient. In addition, they should also do further sensitivity analyses among CKD 4 and 5 patients (or patients with eGFR < 20 ml/min).
Further questions about previous concerns:
Comment 4. They should discuss more the characteristics of individuals who needed an increase of diuretics. (change of blood pressure, ACR, Urine Na/Cr, PH, HCO3-, etc)
Comment 6. I know the differences between 24 h urine Na and spot urine Na/Cr. Although they did not have 24 h urine data, they should at least discuss the limitation of their interpretation.
Author Response
Dear Reviewer,
We appreciate very much the time and effort that you have dedicated to providing your valuable feedback on our manuscript. We are very grateful for your insightful comments on our paper. Please find below our responses to your criticism. All changes within the revised version of our manuscript have been highlighted.
Yours sincerely,
Katarzyna Szczecińska
I know the authors had tried their best to respond to my comments. However, as I have mentioned, the major weakness of their research is the study design of a small sample size across a wide range of CKD stages. I think the response to my comment 3 is not justified and insufficient to support their conclusions. The renal handling of acid-base balance and sodium excretion is significantly limited while eGFR less than 20 ml/min. They should provide detailed data (Cr, eGFR, Urea, Na, NT-proBNP, ACR, Urine Na/Cr, SBP, DBP, MAP, NBPF, PWV, HCO3-, and PH) at baseline and post-intervention among each patient. In addition, they should also do further sensitivity analyses among CKD 4 and 5 patients (or patients with eGFR < 20 ml/min/m2).
Response: Thank you for your comment. We have performed the analysis among the patients with eGFR <20ml/min/1.73m2, please find the results enclosed in the supplementary material. The results did not differ significantly from the presented in the article, as for the patients with eGFR <20ml/min/1.73m2 we have also found a significant increase of pH, HCO3 and a decrease in base excess, an increase in serum total protein, albumin, NT-pro-BNP, CRP, PTH, as well as an increase in erythrocyte count, haemoglobin, haematocrit. There were no significant differences in pulse wave velocity or in systolic and diastolic BP values.
Further questions about previous concerns:
Comment 4. They should discuss more the characteristics of individuals who needed an increase of diuretics. (change of blood pressure, ACR, Urine Na/Cr, PH, HCO3-, etc)
Response: Thank you for your suggestion, we have analyzed the characteristics of individuals who needed an increase of diuretics (3 patients) – point by point, value by value. The main difference that we have found, is higher concentration of NT-proBNP at baseline and at the end of the study - V0 mean value- 7610 pg/ml (6499; 3455 and 12875 pg/ml ), V1 mean value 19 226 pg/ml; (23738; 8507 and 25434 pg/ml respectively). The dose increase, mentioned above, may have been required to sufficiently relieve congestive symptoms associated with diagnosed heart failure.
Comment 6. I know the differences between 24 h urine Na and spot urine Na/Cr. Although they did not have 24 h urine data, they should at least discuss the limitation of their interpretation.
Response: Thank you for your suggestion, we have included the discussion of this limitation in the manuscript. „ As sodium intake is associated with elevated blood pressure and cardiovascular disease, the objective evaluation of consumed sodium is important. The measurement of 24-hour urinary sodium excretion is the preferred method of estimating sodium intakes [69], however, 24-hour collections are expensive and relatively burdensome to individuals. Some data indicate that models based on sodium, potassium, and creatinine concentrations from casual urine specimens may be useful for prediction of 24-hour sodium excretion [70]. We have not observed the changes in sodium excretion after NaHCO3 supplementation but the lack of measurement of 24-hour urinary sodium excretion is a limitation of our study. ”
Reviewer 2 Report
The authors have made appropriate changes in the manuscript.
Author Response
Dear Reviewer,
Thank you for your insight
Yours sincerely,
Katarzyna Szczecińska